# Minimally Invasive Management of Inguinal Lymph Nodes in Penile Cancer: Recent Progress and Remaining Challenges

**DOI:** 10.3390/cancers16172935

**Published:** 2024-08-23

**Authors:** Ahmet Murat Aydin, Emily Biben, Alice Yu, Nicholas H. Chakiryan, Reza Mehrazin, Philippe E. Spiess

**Affiliations:** 1Department of Urology, University of Arkansas for Medical Sciences, Little Rock, AR 72205, USA; 2Department of Genitourinary Oncology, Moffitt Cancer Center, Tampa, FL 33612, USA; 3Department of Urology, Oregon Health & Science University, Portland, OR 97239, USA; 4Department of Urology, Icahn School of Medicine at Mount Sinai, New York, NY 10029, USA

**Keywords:** image-guided surgery, lymph node excision, minimally invasive, nodal staging, penile squamous cell carcinoma

## Abstract

**Simple Summary:**

Penile cancer is an uncommon malignancy that affects less than 1% of men in the Western world. However, it is often aggressive, and the survival of patients is mainly dependent on the presence and spread of nodal metastasis. Penile cancer has an expected pattern of spread to the inguinal lymph nodes in a stepwise fashion; therefore, inguinal lymph node dissection has been an integral part of penile cancer staging and treatment. Minimally invasive inguinal lymph node dissection with the use of laparoscopy and robotics has demonstrated decreased morbidity with successful outcomes. Moreover, sentinel lymph node biopsy has become the preferred method for evaluation of inguinal lymph node metastasis in penile cancer patients with nonpalpable inguinal lymph nodes. Here, we extensively evaluate the recent developments in the management of inguinal lymph nodes in penile cancer, review the advantages and shortcomings of the available treatment strategies, and discuss the remaining challenges and future perspectives.

**Abstract:**

The diagnosis of occult inguinal lymph node metastasis in clinically node-negative invasive penile squamous cell carcinoma (PSCC) has remained a challenge, with substantial perioperative complications. The recent refinements in the technique of dynamic sentinel lymph node biopsy (DSLNB) demonstrated high diagnostic accuracy with considerably lower morbidity compared to conventional open modified/superficial inguinal lymph node dissection (ILND). Although DSLNB, if available, has been endorsed as the preferred method for nodal staging in patients with invasive PSCC and no palpable inguinal lymphadenopathy in the recent penile cancer guidelines, its utilization has been quite limited so far. Laparoscopic and robotic-assisted ILND have emerged as alternatives for nodal staging in this patient population and are shown to improve the rate of wound infections and postoperative pain. For management of nodal metastasis in patients with clinically palpable inguinal lymph nodes, minimally invasive ILND has shown promising results as well. Nonetheless, given the rarity of PSCC and the absence of prospective studies and clinical trials, nodal staging and treatment of nodal metastasis in clinical practice will likely continue to vary across the medical centers in the following years. In this review, we first summarize the evolution of DSLNB and minimally invasive ILND and discuss the advantages and drawbacks of each management strategy. We further discuss the remaining challenges and future perspectives in the management of inguinal lymph nodes in patients with PSCC.

## 1. Introduction

Approximately 25% of patients with cN0 disease have micrometastatic disease that cannot be detected by conventional imaging alone [1,2]. The presence of lymphovascular invasion, greater than 50% poorly differentiated cancer and high pathological tumor stage (pT1G3-4 or any ≥pT2) are significant predictors of lymph node metastasis and are defined as “high-risk features” by the current NCCN guidelines [1,3]. Compared to delayed ILND, the early prophylactic removal of occult metastasis in inguinal nodes was shown to improve survival in patients with PSCC [4]. Ideally, this survival benefit from early detection and surgical removal of nodal metastasis in high-risk patients should outweigh the harm from procedure-related morbidity [5].

Traditionally, surgical comorbidity has been reported in 60–100% of patients undergoing ILND [6,7]. Lymphedema (16–50%), wound dehiscence (11–50%), lymphocele (9–15%), infection (9–14%), and skin necrosis (0–8%) are common complications [6]. Rare complications include deep vein thrombosis and pulmonary thromboembolism (4–6%), vascular injury and hematoma (2–4%), neuropraxia, and neuronal injury (2%). Thanks to the technical modifications of ILND and the use of superficial/modified ILND for staging, complication rates have improved, though they are still high (19 and 27% rate of minor and major complications, respectively). The high morbidity has remained a great concern in clinical practice; an analysis of the Surveillance, Epidemiology, and End Results database demonstrated that in patients who showed indication of curative ILND, only 25% underwent the procedure [8]. Several other attempts to refine the technique and reduce ILND-associated lymphorrhea such as use of tissue sealant, ultrasonic scalpel, and intraoperative lymphatic mapping with peri-incisional methylene blue injection showed little to no success or limited practicality [9,10,11]. Thus, recently, the role of minimally invasive surgery of inguinal nodes in PSCC has become of research interest and evaluated in several medical centers. Although there is still limited information on minimally invasive ILND, it has been shown to be safe and effective, with lower morbidity than open surgery in carefully selected patient populations with nonpalpable or non-bulky inguinal nodes [12]. Remarkably, the lymphedema rate was found to be three times less common in minimally invasive ILND compared to open ILND.

Sentinel lymph node biopsy is an alternative surgery for patients with PSCC, and its potential value has been actively explored for several decades [13]. Particularly, dynamic sentinel lymph node biopsy (DSLNB) has emerged as a promising modality, thanks to the ongoing refinements in the technique and improvements in false-negative rates. Nonetheless, technical complexity requiring a highly experienced multidisciplinary team currently remains as a barrier against its wider utilization for lymph node staging in the current era.

The current NCCN guideline panel recommends DSLNB as an alternative lymph node staging option to bilateral modified inguinal lymph node dissection (ILND) in patients with ≥T1G3 disease and no palpable/visible (cN0) inguinal lymph nodes if the treating physician has experience with this modality [3]. Moreover, DSLNB is now advocated by the EAU-ASCO collaborative panel as the preferred method for lymph nodal staging in high-risk nonpalpable penile cancer [14]. The former has not included minimally invasive ILND in its recommended treatment algorithms, whereas the latter recommended use of minimally invasive ILND as part of a clinical trial only. In this review, we first summarize the evolution of DSLNB and minimally invasive ILND and discuss the advantages and drawbacks of each management strategy. We further discuss the remaining challenges and future perspectives in the management of inguinal lymph nodes in patients with PSCC.

## 2. Materials and Methods

This narrative review aims to evaluate the recent progress in minimally invasive management of inguinal lymph nodes in penile cancer and the advantages and drawbacks of each management strategy. Two authors of this study performed a comprehensive review across electronic databases using PubMed, EMBASE, and the Cochrane Database of Systematic Reviews, by using a combination of MeSH (medical subject heading) terms. The exact MeSH terms utilized for the database search were penile neoplasms, lymphatic metastasis, lymph node excision, sentinel lymph node biopsy, and minimally invasive surgical procedures. We further narrowed down our search results based on predefined terms related to penile cancer, dynamic sentinel lymph node biopsy, and video endoscopic (laparoscopic and robotic) inguinal lymph node dissection. Specifically, we subdivided minimally invasive management into two categories: dynamic sentinel lymph node biopsy and video endoscopic inguinal lymphadenectomy. To include other pertinent studies, we also conducted a manual review of recent review articles, ongoing clinical trials, and recently published scientific abstracts available in the literature. We also excluded studies not published in English and studies published prior to 2001 (Figure 1).

## 3. Results

### 3.1. DSLNB in Nodal Staging of Penile Cancer

Sampling sentinel lymph nodes, the first echelon of lymph nodes that receive lymphatic drainage from a peritumoral region, was first described in 1977 [15]. The technique of sentinel lymph biopsy for PSCC has constantly evolved during the last five decades, and particularly within the last few years, significant improvements have been achieved [13] (Figure 2). Dell’Oglio et al. from the Netherlands Cancer Institute refined their technique of DSLNB and reported their outcomes with a hybrid fluorescent–radioactive tracer, using indocyanine green (ICG)-99mTc-nanocolloid in a combined radio- and fluorescence-guided sentinel node biopsy for PSCC [16]. The false-negative probability for a negative procedure was only 1.4% at a two-year follow-up, outperforming previously described sentinel lymph node biopsy techniques in PSCC. The procedure was considerably well tolerated and safe, with a reported overall and major postoperative complication rate of 22% and 3%, respectively. The same group also evaluated short-term postoperative complication rates among 644 patients treated with the same DSLNB technique [17]. They reported that 80% of all 90-day complications occurred in 30 days. The 30-day postoperative complication rate was about 15% (per groin), of which 30% and 64% were Grade I and Grade II complications per the Clavien–Dindo classification. Wound infections (10%) and lymphoceles (3%) were the most common complications occurring within 30 days of DSLNB. In 0.5% of cases, surgical intervention was required due to infected lymphocele or postoperative bleeding. Three patients had postoperative atrial fibrillation, sepsis, and hemodynamic instability due to infection and postoperative bleeding and were admitted to the intensive care unit within 30 days of DSLNB. The 30- and 90-day postoperative complication rate was about 3.5%, none of which were Grade III complications or higher.

Interestingly, the number of removed lymph nodes per groin was the main independent predictor for any 30-day complications and Grade ≥ II complications [17], thus there appears to be a fine balance between improving staging accuracy and minimizing postoperative complications in DSLNB. Although the postoperative complication rates in DSLNB appeared to be significantly lower than those occurring after ILND, it is still considerably high. Therefore, further research and refinement in this DSLNB technique are warranted to achieve more accurate detection of the true sentinel lymph node and lower comorbidity.

Likewise, O’Brien et al. from Australia recently reported successful outcomes of DSLNB [18]. In addition to preoperative lymphoscintigraphy with single-photon emission computed tomography (SPECT/CT) and intraoperative gamma-probing, they utilized blue dye injection. Among 64 patients (127 groins) who underwent DSLNB between 2015 and 2021, the false-negative rate was only 1.9% and the sensitivity of DSLNB was 90.5%. DSLNB avoided prophylactic radical ILND in 71.7% of groins and allowed the patients to proceed with surveillance instead. Nevertheless, despite these promising false-negative rates reported, DSLNB is a technically challenging procedure, and the aforementioned near-excellent outcomes could not be reproduced in other studies [19].

In an another defined DSLNB technique, ultrasonography (US)-guided fine-needle aspiration (FNA) was utilized. Instead of implementing preoperative SPECT/CT and fluorescence guidance using a near-infrared camera during DSLNB, Lee et al. evaluated all groins with US and then performed DSLNB ± US-guided lymph node excision following initial negative US-guided FNA among 403 men (728 groins) [20]. The sensitivity value and negative predictive value of this simplified DSLNB technique were 96% and 100%, respectively. The authors suggested that US-guided FNA prior to DSLNB also brings logistic advantages, since synchronous ILND and pelvic lymph node dissection can be performed if initial FNA detects two or more positive lymph nodes. Moreover, compared to upfront evaluation with DSLNB, US-guided FNA prevents scarring which might interfere with dissection planes in subsequently indicated minimally invasive ILND.

Recently, in a pilot study (SentiPen), the feasibility of subcutaneous injection of superparamagnetic iron oxide particles (Sienna+/SentiMag technique) was evaluated for detecting inguinal sentinel nodes in patients with cN0 PSCC [21]. The cancer detection rate and median number of nodes excised per inguinal region was similar between standard radioisotope-guided DSLNB and DSLNB using superparamagnetic iron oxide particles (Sienna) and its probe detecting magnetic signature (SentiMag probe). This novel approach eliminates the need for lymphoscintigraphy and the use of blue dye (and thus the risk of anaphylaxis and blue dye tattooing), reduces the exposure to radioactivity, and allows earlier administration of superparamagnetic particles in advance, i.e., 7 days before surgery.

Thanks to the recent promising outcomes reported from the refined DSLNB techniques, the feasibility of avoiding completion ILND following a positive DSLNB was also explored in a retrospective multi-center European cohort [22]. A prediction model that included the number of positive inguinal nodes in a positive DSLNB and the largest nodal metastasis size as predictive parameters failed to determine inguinal basins in which ILND could be safely omitted. About 15% of inguinal basins with a positive DSLNB harbored additional lymph node metastases at the subsequently performed completion ILND. For the time being, completion ILND remains the standard of care for all inguinal basins with a positive DSLNB.

### 3.2. Minimally Invasive ILND for Nodal Staging of Penile Cancer

The first endoscopic groin dissection for inguinal lymphadenopathy in penile cancer patients was described by Bishoff et al. on two human cadavers and one patient in 2003 [23]. The technique, however, was unsuccessful in the living patient whose dissection was converted to open due to fixed, enlarged lymph nodes to the femoral vein which were not amenable to video endoscopic dissection [24]. Nonetheless, in parallel to the increasing utilization of minimally invasive surgery for urological oncology procedures over the years, the value of minimally invasive inguinal lymph node dissection has been continuously explored for both diagnostic and therapeutic indications [25]. The video endoscopic (laparoscopic) inguinal lymphadenectomy (VEIL), which can also be performed with robotic assistance (Figure 3), have been tested for lymph nodal staging in high risk, clinically node-negative populations. It has been suggested that nodal staging with minimal invasive ILND might decrease surgical comorbidity compared to open modified ILND and could also eliminate the risk of non-visualization of sentinel nodes during DSLNB [26].

Tobias-Machado et al. first described VEIL in 2006 [24]. They reported decreased operative times, reduced hospital stays, earlier returns to activities, and fewer complications. Cutaneous and lymphatic complications, and hematomas totaled to 20% in the VEIL subset, while 70% experienced complications from conventional open surgery. Sudhir et al. suggested similar findings in 22 patients (19 patients with penile cancer, 2 patients with vulva cancer, and 1 with urethral cancer) who underwent VEIL surgeries with decreased risk of skin necrosis and less pain, though their sample was not directly compared to open surgery [27]. Skin complications are likely decreased due to port placement and incision in the thigh rather than the groin. Similarly, Master et al. discussed endoscopic groin dissection in several malignancies including penile, melanoma, and gynecologic, and reported only 14.6% major complications in 41 groin dissections, with most wound infections requiring escalation of antibiotic therapy only [28]. No cases of mortality or debilitating lymphedema were reported in these cases. Nonetheless, recent data reported no significant advantage of the VEIL approach in preventing lymphocele compared to the open approach [29]. A recent systematic review and meta-analysis demonstrated that the incidence of lymphatic-related complications varied between 20 and 50% and was comparable among patients who underwent open ILND or VEIL [30]. It can be argued that the number of inguinal lymph nodes rather than the surgical approach is the main driving factor in the development of lymphoceles, which appears to underscore the unique benefit obtained from DSLNB.

However, one of the concerns with DSLNB is non-visualization of sentinel nodes due to unfavorable chemical characteristics of ICG, which rapidly diffuses through the lymph nodes resulting a fluorescent node that is not necessarily the true sentinel lymph nodes, and/or obscurement of another fluorescent node by overlying fat due to limited tissue penetration depth of the fluorescent signal [26]. Hora et al. recently tested a novel technique in a pilot study of men with ≥pT1G2 cN0 PSCC, a combination of VEIL with fluorescent ICG, which enabled both florescence marking of sentinel lymph nodes and direct endoscopic visualization of sentinel lymph nodes [26]. Although concomitant use of fluorescence infrared image with ICG during VEIL was proven to be technically feasible and shown to improve the probability of removing the true sentinel lymph node, a considerably high grade I-II lymphocele rate (40%) was reported in addition to the high procedural cost.

Matin et al. recently performed a prospective phase I trial among 10 patients with T1-3N0 PSCC to evaluate the adequacy of robotic-assisted VEIL [31]. Per protocol, after the completion of robotic-assisted VEIL, an open ILND was performed at the same site by an independent surgeon via a separate open incision. Each side of the dissection took between 90 and 120 min, the median estimated blood loss was 100 mL, and the mean numbers of superficial inguinal nodes removed was 9 (range 5–21) with no intraoperative vascular or neurological injuries. Adequate dissection was achieved in 18 of 19 inguinal fields, suggesting that robotic-assisted VEIL allowed adequate staging of nodal disease in inguinal regions among patients with high-risk localized PSCC.

Almost all reports from the recent VEIL studies showed promising results; however, VEIL is still in its early stages. An online questionnaire for capturing the variations in PSCC care across different regions of the world performed by Global Society of Rare Genitourinary Tumors (GSRGT) demonstrated that VEIL was the least commonly utilized method for nodal staging (14%) among all options, particularly when compared to modified/superficial ILND (58%), which was the most utilized [32]. Due to its near-excellent detection rates and low comorbidity, DSLNB will likely have precedence over VEIL in the staging of nodal disease at high-volume centers with the availability to offer it. Nonetheless, in several countries without centralization of penile cancer care, the availability and quality of DSLNB will likely remain an issue; thus, VEIL may be more commonly preferred over open ILND in the staging of nodal disease. Further studies with larger multi-center cohorts and longer follow-ups are needed to evaluate the value of VEIL for this indication.

### 3.3. Minimally Invasive Management of Nodal Metastasis in Penile Cancer

Unlike its infrequent use for nodal staging, the most common indication that VEIL has been utilized in the management of PSCC is the treatment of nodal metastasis. Most authors have tested lymph node yield and postprocedural nodal recurrence as the measures of oncologic efficacy for lymph node dissection (Table 1). Singh et al. compared lymph node yield in open versus robotic ILND and demonstrated that the median inguinal lymph node count per inguinal basin was comparable (approximately 13 lymph nodes) in both approaches [33]. No patient in their cohort experienced recurrence after a median follow-up of 40 months, suggesting the oncological comparable efficacy of the robotic technique. This is further reflected in Thyavihally et al. who retrospectively compared long-term survival outcomes in a heterogenous group of penile cancer patients who underwent both diagnostic and radical ILND, via open and minimally invasive approaches [29]. Recurrence rates (inguinal and distant) were consistent among patients who underwent O-ILND (28%) and VEIL (25%). Overall survival was 8 months longer in the VEIL group, while five-year survival was comparable. Unlike type of surgery, statistically significant factors for predicting survival were grade of tumor, pathological nodal status, and partial versus total initial treatment for the primary disease [29].

With the increasing use of robotic surgery for ILND, more data has become available, enabling comparisons of robotic-VEIL versus open ILND in terms of lymphocele incidence and operative times. Bulky or fixed lymph nodes were considered not amenable to robotic dissection. However, among PSCC patients who were amenable to robotic-VEIL, Singh et al. reported no incidence of severe lymphedema in patients who underwent robotic-VEIL in contrast to 9% of patients who underwent open surgery developing severe lymphedema [33]. The authors suggested that lymphatics might have been better preserved with insufflation in robotic surgery that might assist in an atraumatic retraction rather than mechanical retractors used in open surgery. The authors also speculated that smaller port incisions in the thigh instead of bigger incisions in the groin might further contribute to better preservation of skin lymphatics and vasculature, which might explain the fewer lymph- and skin-related complications in the robotic surgery group. Their findings, however, contrasted with recently reported outcomes from the other studies which showed no significant advantage of VEIL in preventing lymphocele formation compared to the open approach [29,34].
cancers-16-02935-t001_Table 1Table 1A summary of reported perioperative outcomes from the comparative studies of open ILND and VEIL in patients with PSCC.StudyInterventionNumber of PatientsClinical Nodal StageLymph Node Yield/GroinNodal Recurrence/GroinOperative Time (min)Hospital Stay (days)LymphoceleInfectionSkin Dehiscence and NecrosisSingh et al., 2018 [33]Open ILND10061% N024% N115% N212.5 (10.5–14.3)None(med. follow-up: 40 months)60 (55–70)4 (3–5)55%17%36%VEIL5166.7% N019.6% N113.7% N213 (11–14.5)None (med. follow-up: 41 months)75 (70–84.8)3 (3–3.8)49%7.8%11%Thyavihally et al., 2021 [29] ^1^Open ILND3221.9% N050% N128.1% N210.5 (7–16)9.3%(med. follow-up: 51 months)110 (70–190)9.6 (5–20)23.7%42.3%23.7%VEIL4738.3% N044.7% N117% N210 (7–18)4.2%(med. follow-up: 42 months)90 (50–140)6.1 (4–12)20.4%8%0%Shao et al., 2022 [35]Open ILND6930% N020% N150% N27.39 ± 3.9Not reported65  ± 29.413.9  ± 10.0913%6.5%2%VEIL4017% N020% N163% N26.7 ± 3.6Not reported60  ± 22.88.9  ± 7.913.8%2.5%1%Yadav et al.,2018 [34] ^2^Open ILND2976% N014% N110% N28.3 (8–9)Not reported92 (74–120)10.2 (14–22)13.7%13.8%27%VEIL2973% N017% N110% N27.6 (7–8)Not reported163 (132–195)4.7 (4–8)10.3%0%6.8%^1^ In the open ILND group, 27 patients underwent bilateral dissection and 5 underwent unilateral dissection. In the VEIL group, 41 patients underwent bilateral dissection and 6 underwent unilateral dissection. ^2^ A total of 29 patients included in this study underwent both open ILND and VEIL, with one groin treated with open ILND and the contralateral side treated with VEIL.

The other main concerns regarding VEIL were possible prolonged procedure times and increased procedure costs. Operative time in the robotic approach (70 to 85 min) was often found to be longer than that in the open approach (55–70 min), which included port placement and docking time; however, other reported VEIL times, which ranged from 91 to 165 min, were notably longer than robotic-VEIL [33]. A Chinese study recently reported the feasibility of simultaneous double radical VEIL for bilateral inguinal nodes in 65 PSCC patients with cN1 and cN2 disease. Patients who underwent double concomitant radical VEIL (22 patients) had significantly shorter operative times (105 ± 10 min) than those who underwent sequential open bilateral radical ILND (19 patients, 160 ± 13 min) and sequential bilateral radical VEIL (24 patients, 191 ± 17 min, *p* < 0.001) [36]. The patients who underwent double and single radical VEIL were noted to have similarly decreased complication rates compared to open radical ILND (5% each vs. 63%), as well as shorter postoperative hospitalization (9 days each vs. 13 days), lower incidence of wound infection (1 patient each vs. 8 patients), skin necrosis (0 each vs. 7 patients), and lymphorrhea (1 patient each vs. 8 patients). Furthermore, Tobias-Machado et al. reported the feasibility of performing simultaneous bilateral VEIL and bilateral pelvic lymphadenectomy in a patient with pT3cN2 PSCC with acceptable outcomes [37]. They recommended this approach for patients at a greater risk for pelvic lymph node disease metastasis, noncompliant patients that may miss subsequent surgeries and patients having difficulty in accessing health services, since it enables a single-shot nodal treatment and potentially earlier application of adjuvant treatments.

Although several previous retrospective studies have reported consistent findings suggesting that VEIL could provide similar oncologic efficacy compared to open ILND with less postoperative complications, notably wound infections and skin necrosis [35,38,39], the added benefit of minimally invasive ILND in terms of healthcare spending was not justified yet. Indeed, several minimally invasive thoracic and abdominopelvic oncological procedures, particularly radical prostatectomy, cystectomy, and nephrectomy, were shown to be cost-effective or potentially cost-effective compared to comparative open procedures [40]. The main drivers for increased cost were length of hospital stay and equipment costs. In previous studies of ILND for PSCC, the procedural costs appeared to be higher in robotic procedures; however, hospital stay, time off work, postoperative re-admissions, and costs of complications were not considered [33]. Taking the multitude of additional variables into account may likely give better insight to the cost of different ILND approaches. Of these variables, it is worth mentioning that in one study, the length of hospital stay, as expected, was significantly shorter, by 5 days, in patients who underwent minimally invasive ILND compared to those who underwent open surgery [41]. Finally, a few recent case reports also demonstrated the feasibility and safety of single-port robotic-ILND (Figure 4) for the treatment of palpable inguinal lymph nodes in patients with PSCC [42,43,44].

Despite the available retrospective studies of ILND included heterogenous groups of penile cancer patients and the lack of randomized controlled prospective studies comparing therapeutic ILND approaches, there appears to be sufficient data supporting the use of minimally invasive therapeutic ILND with VEIL and robotic-VEIL for the management of nodal metastasis in PSCC. The growing field of minimally invasive surgery and its wider utilization and adaptation by surgeons will likely lead to increased utilization of radical VEIL in penile cancer. Of note, patients now preferentially choose the endoscopic minimally invasive surgical approach to the open approach whenever offered both options, given the decreased surgical morbidity rates in VEIL [45]. In this regard, it is suggested that conducting a non-inferiority clinical trial comparing minimally invasive to open ILND may not be feasible.

## 4. Future Perspectives

Currently, there are several challenges that need to be addressed in the management of inguinal nodes in PSCC. Firstly, there is more standardized inguinal node management in PSCC worldwide. A recent European online survey has demonstrated that the technique of radical ILND and DSLNB performed by high-volume surgeons significantly differed in almost every aspect, from dye injection site, use of lymphoscintigraphy, and techniques for lymphatic control, to duration of empiric antibiotic therapy, scheme of perioperative thromboprophylaxis, and time points for drain removal [46]. In addition to variations in the preferred procedural technique and perioperative management among physicians, the significant heterogeneity of patient populations in previous studies has been the other major confounding factor that prevented us from making reliable comparisons among DSLNB, VEIL, and open ILND.

Centralization of PSCC care, particularly in advanced settings, appeared to benefit patients since proper evaluation of inguinal lymph nodes in PSCC was often neglected in a decentralized setting [47]. In this regard, the centralization of penile cancer care and referral to high-volume medical centers may improve the utilization rate of DSLNB. Timely referral of patients to high-volume centers performing DSLNB will be crucial in preventing treatment delays since delays in ILND of more than 3 months are associated with poorer 5-year recurrence-free survival outcomes (77% vs. 38%) [48]. Moreover, DSLNB can be possibly utilized for newer indications. Nodal metastasis can be detected in roughly 5% of patients with low-risk localized PSCC (pTa, pT1G1), and whether the 5% risk of metastasis in a deadly disease is a small risk or not was previously questioned [5,49]. Due to its low comorbidity and recently demonstrated acceptable detection rate, it has been postulated that DSLNB might be utilized as a new assessment tool for nodal metastasis risk in patients with low-risk PSCC [5]. In a recent single-center Italian study that included 93 low-risk PSCC (pT1aG1 cN0M0) patients, 7 (7.5%) out of 93 patients were found to have inguinal recurrence after a median interval of 9 months following primary tumor surgery [50]. The majority of the patients with recurrence was younger males, and it was suggested that young men with low-risk PSCC might be offered minimally invasive staging of lymph nodes after primary tumor surgery, such as DSLNB, as an alternative to observation alone.

While the treatment landscape of inguinal lymph nodes in PSCC quickly evolves, there are currently a few ongoing clinical studies which may shed more light into the management of inguinal lymph nodes in PSCC. The VELRAD study from the UK was designed to assess the feasibility of performing a randomized clinical trial comparing radical VEIL and radical open ILND (NCT05592639) [51]. The trial has a target enrollment of 50 patients with PSCC and will test these two procedures in a variety of indications: for patients with small-volume palpable inguinal lymph nodes (<2 cm on CT) not fixed to skin, nodal staging in patients with PSCC or mucosal melanoma of the penis > T1bG2, patients with previous DSNB with confirmed metastatic inguinal nodes requiring a completion radical inguinal lymphadenectomy and patients with impalpable nodes not suitable for DSNB. Patients with palpable inguinal lymph nodes fixed to skin or adjacent structures will be excluded. Another single-center prospective observational study will assess the patient experience and needs for patients undergoing DSLNB and/or ILND for PSCC using decision regret scale questionnaires (NCT05842031) [52]. It aims to identify what interventions may be helpful in improving quality of life, patient educational needs, and symptom management. While the Inpact trial is the only ongoing phase III clinical trial being conducted for PSCC and expected to inform optimal sequencing of radical ILND in advanced PSCC management, it is not designed to compare minimally invasive radical ILND to open ILND (NCT02305654) [53]. Nonetheless, as a secondary study outcome, it will give a high level of evidence for operability, which will be recorded as whether the planned ILND was undertaken and the reasons if it did not occur, and incidence rates of lower limb/scrotal edema.

## 5. Conclusions

The emergence of minimally invasive management options for inguinal nodal management of PSCC and recent refinements in techniques of nodal staging and nodal metastasis treatment have improved treatment-related morbidity without compromising oncological efficacy. Particularly, compared to open superficial/modified ILND, the rate of postprocedural complications such as surgical-site infection and lymphocele was lower in VEIL but still higher than those reported in DSLNB. Moving forward, DSLNB may replace ILND as the standard modality of nodal staging in nonpalpable cN0 PSCC in high-volume medical centers. For the treatment of non-bulky palpable cN+ disease, particularly palpable inguinal lymph nodes smaller than 2 cm in diameter, radical VEIL appears to cause less comorbidity than open ILND and may eventually be more commonly utilized in this setting. Open radical ILND will remain as an important procedure and the mainstay treatment in patients who have bulky inguinal nodal metastasis with large and fixed palpable inguinal nodes.

## Figures and Tables

**Figure 1 cancers-16-02935-f001:**
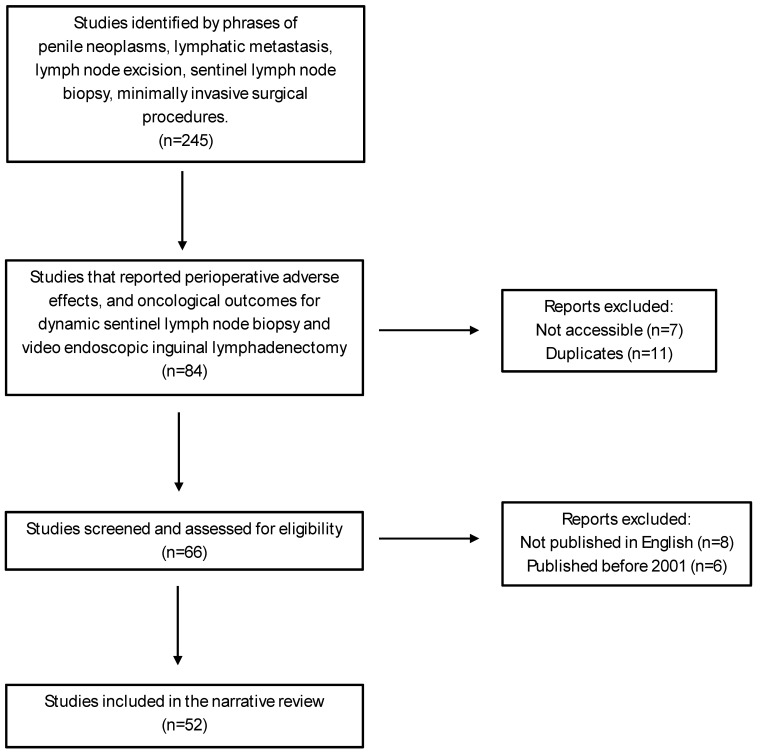
Summary of literature search.

**Figure 2 cancers-16-02935-f002:**
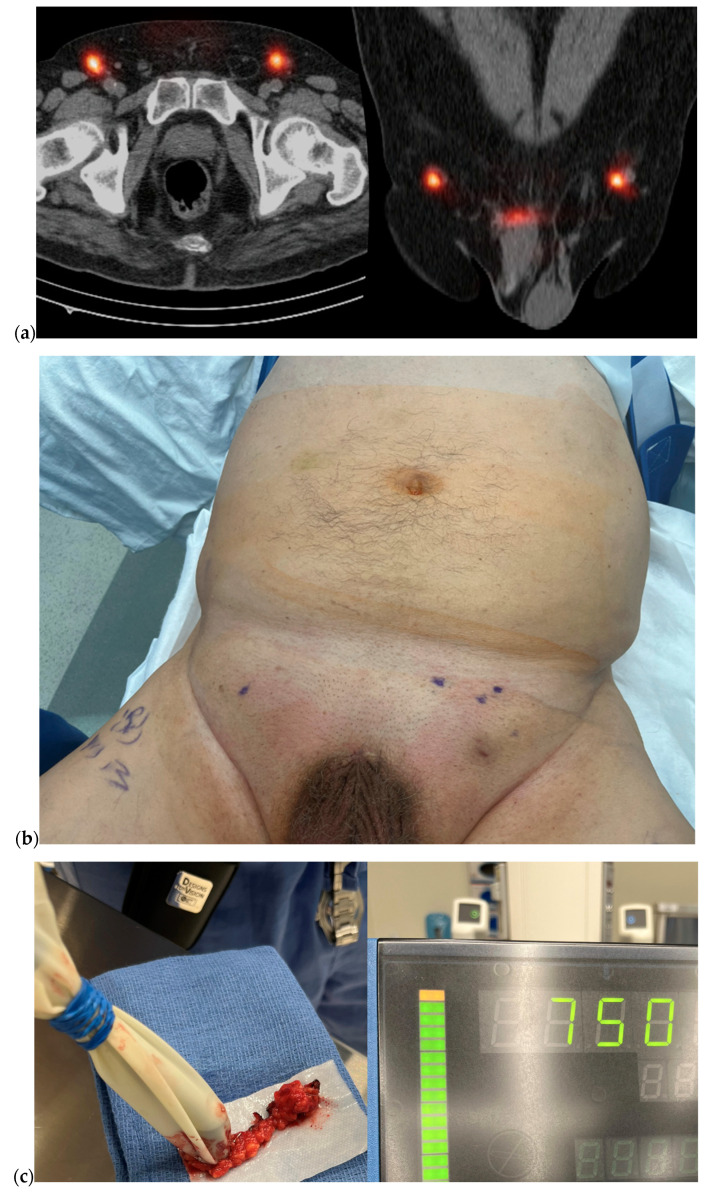
An example PSCC case who underwent DSLNB for staging of inguinal lymph nodes. (**a**) Preoperative axial and coronal images of single-photon emission computed tomography/computed tomography (SPECT/CT) demonstrating sentinel lymph nodes in bilateral groins. (**b**) Skin markings showing the location of sentinel lymph nodes where the incisions were to be made. (**c**) Intraoperative image of gamma-probe re-confirming high radioactivity in the resected inguinal lymph node.

**Figure 3 cancers-16-02935-f003:**
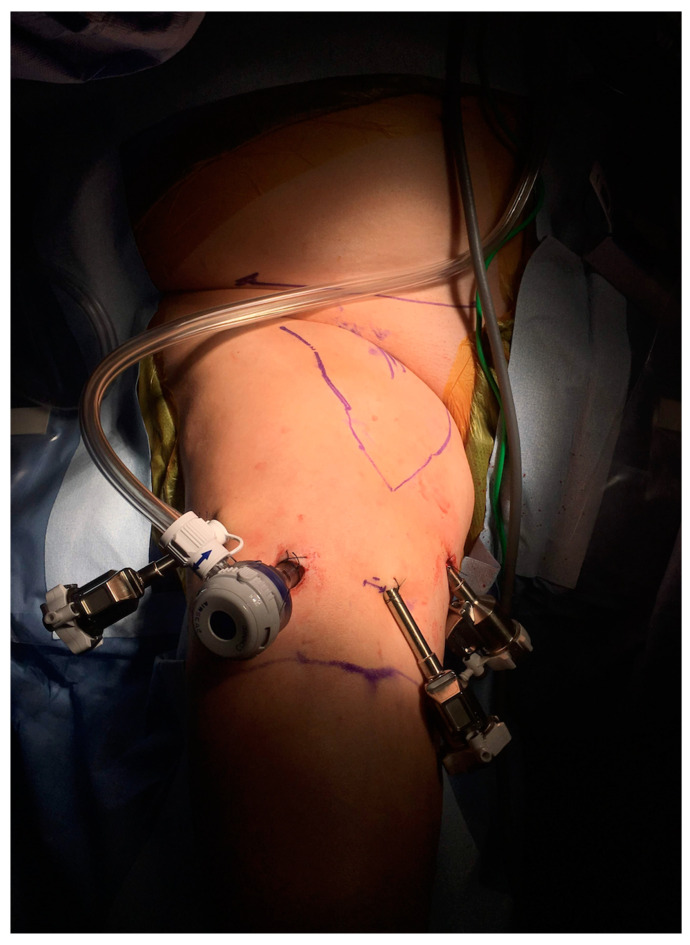
An example PSCC case who underwent multiple-port robotic-VEIL with an intraoperative image showing the placement of robotic trocars.

**Figure 4 cancers-16-02935-f004:**
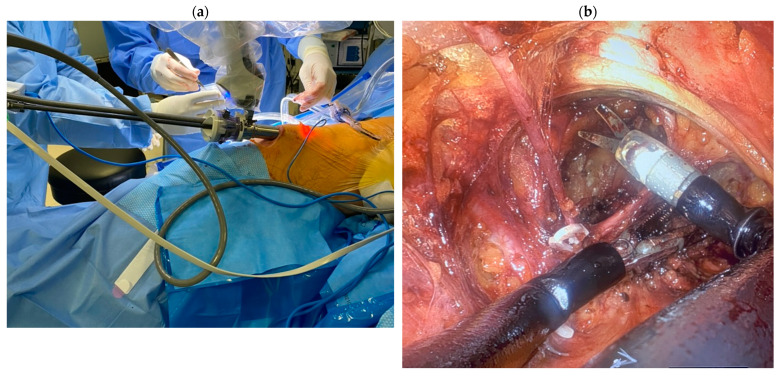
An example PSCC case who underwent single-port robotic-VEIL. (**a**) Placement of a single robotic trocar above the knee. (**b**) Laparoscopic view of the inguinal region during surgical dissection.

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
