# Peer review of "Minimally Invasive Management of Inguinal Lymph Nodes in Penile Cancer: Recent Progress and Remaining Challenges"

_cancers, 2024, doi:10.3390/cancers16172935_

Round 1

Reviewer 1 Report

Comments and Suggestions for Authors

Authors conducted a narrative review for minimally invasive management of node disease in penile cancer. Manuscript is well written and comprehensive, well organized and there are a few clear and informative images. I believe most updated literature is discussed and therefore authors should be commended on that. I would suggest authors to add methodology section, since this is clearly not a systematic review but still a short comment on methodology should be made.

1. What is the main question addressed by the research? To summarize evidence regarding minimally invasive options for management of lymph node metastases in patients with penile cancer.
2. What parts do you consider original or relevant for the field? What 
specific gap in the field does the paper address?  This is a systematic review providing updated evidence of the literature on minimally invasive management of lymph node metastases in patients with penile cancer.
3. What does it add to the subject area compared with other published 
material? Updated systematic review of the literature.
4. What specific improvements should the authors consider regarding the 
methodology? What further controls should be considered? Methodology section should be added to include what search algorithm was used, which databases were searched and up to which specific date.
5. Please describe how the conclusions are or are not consistent with the 
evidence and arguments presented. Please also indicate if all main questions 
posed were addressed and by which specific experiments. Conclusions were in line with the presented literature and answered by the data provided for all types of diagnostic modules presented.
6. Are the references appropriate? Yes
7. Please include any additional comments on the tables and figures and 
quality of the data. Figures should be cited appropriately if taken from another source.

Author Response

ANSWER: We thank the author for his great comments. The clinical photos presented as figures in the manuscript were all captured by the authors of this review article. They were not taken from another source that would require any citation.

As the reviewer suggested, we incorporated a methodology section in the revised manuscript as follows:

  1. Materials and Methods.
    This narrative review aims to evaluate evolution of minimally invasive management of inguinal lymph nodes in penile cancer, and advantages and drawbacks of each management strategy. Two authors of this study performed a comprehensive review across electronic databases using PubMed, EMBASE, and the Cochrane Database of Systematic Reviews, by using a combination of MeSH (medical subject headings) terms with no language restrictions. Our search employed predefined terms related to penile cancer, inguinal lymph node metastasis, sentinel lymph node biopsy, inguinal lymph node dissection, and minimally invasive surgery. To include other pertinent studies, we also conducted a manual review of recent review articles, ongoing clinical trials, and recently published abstracts available in the literature.

Reviewer 2 Report

Comments and Suggestions for Authors

This narrative review is comprehensive even though a literature search strategy hasn't been presented and this should be revised. 

1. What is the main question addressed by the research? The review concerns the minimal invasive technique for inguinal lymph node clearence. However this is a narrative review, without any literature search strategy
2. What parts do you consider original or relevant for the field? What
specific gap in the field does the paper address? The paper only highlights the advantages of the minimal invasice technique buy citing favourable literature.
3. What does it add to the subject area compared with other published
material? This narrative review does not bring anything new to the field.
4. What specific improvements should the authors consider regarding the
methodology? What further controls should be considered? At this level of impact (high impact journal) this should be at least a systematic review.  
5. Please describe how the conclusions are or are not consistent with the
evidence and arguments presented. Please also indicate if all main questions
posed were addressed and by which specific experiments. The evidence is selectively presented to support the evidence.  
6. Are the references appropriate? Yes  
7. Please include any additional comments on the tables and figures and
quality of the data.  

Author Response

ANSWER: We thank the author for his comments and suggestions. He also suggested a methodology section for this manuscript as reviewer 1, thus we agreed to incorporate a methodology section in the revised manuscript. Of note, we also explained our research strategy in this new subsection as follows:

“2. Materials and Methods.
This narrative review aims to evaluate evolution of minimally invasive management of inguinal lymph nodes in penile cancer, and advantages and drawbacks of each management strategy. Two authors of this study performed a comprehensive review across electronic databases using PubMed, EMBASE, and the Cochrane Database of Systematic Reviews, by using a combination of MeSH (medical subject headings) terms with no language restrictions. Our search employed predefined terms related to penile cancer, inguinal lymph node metastasis, sentinel lymph node biopsy, inguinal lymph node dissection, and minimally invasive surgery. To include other pertinent studies, we also conducted a manual review of recent review articles, ongoing clinical trials, and recently published abstracts available in the literature.”

Nevertheless, we certainly believe that this area, minimal invasive management of inguinal lymph nodes in penile cancer, requires a comprehensive update currently. And the remaining challenges and future perspective should be discussed in detail by the penile cancer experts based on recent accumulating data since there is currently a paucity of information on this topic, there is no comprehensive study comparing conventional and novel minimally invasive techniques, in spite of the fact that minimally invasive procedures for inguinal nodes are being more commonly utilized in current practice and there is significant heterogeneity in terms of practice patterns even between academic medical centers. With this review, we aim to address this gap as much as possible.

Reviewer 3 Report

Comments and Suggestions for Authors

The authors present recent developments in dynamic sentinel lymph node biopsy (DSLNB) in invasive penile squamous cell carcinoma (PSCC) compared to conventional open modified/superficial inguinal lymph node dissection (ILND). The trends and future prospects of laparoscopic and robot-assisted ILND are also discussed, although there is not enough established evidence.

Basically, the argument is clear and the supporting literature is properly cited.

However, minor modifications are needed on the following points.

Minor points:

#1. Regarding Figure 3(b), a captured image of robot-assisted ILND is presented as an intraoperative finding. However, this image alone is not clear to the reader, so please illustrate a simple structure that will serve as a merkmal.

#2. Author contributions and Funding have not been completed. Please fill them in appropriately.

Author Response

ANSWER: We thank the author for his excellent comments. We included author contributions and funding date in the revised manuscript as follows:

“Author Contributions:
Conceptualization, A.M.A., P.E.S.; writing—original draft preparation, A.M.A., E.B.; writing—review and editing, N.H.C., A.Y., R.M., P.E.S.; supervision, A.M.A., N.H.C., A.Y., R.M., P.E.S.; project administration, A.M.A. All authors have read and agreed to the published version of the manuscript.

Funding: This research received no external funding.”

We also thank the author bringing Figure 3b into our attention. This intraoperative photo in figure 3b does not include all anatomical landmarks in an inguinal lymph node dissection and some anatomical landmarks are not clearly depictable either. Moreover, our goal in this figure was to give an idea of intraoperative laparoscopic view in single-port robotic surgery, but not to teach the readers about anatomical structures or surgical steps of inguinal lymph node dissection, which was not our goal but indeed beyond the scope of this review. Therefore, we decided to omit figure 3b from the revised manuscript and only include the Figure 3a as the new figure 3.

Round 2

Reviewer 2 Report

Comments and Suggestions for Authors

The authors have addressed the search strategy issue, but not very well - no mesh terms have been named, no PRISMA flowchart has been completed. 

Author Response

We thank the reviewer for his feedback. We included a new figure to summarize our search strategy. We also revised methodology as follows:

This narrative review aims to evaluate recent progress in minimally invasive management of inguinal lymph nodes in penile cancer, and advantages and drawbacks of each management strategy. Two authors of this study performed a comprehensive review across electronic databases using PubMed, EMBASE, and the Cochrane Database of Systematic Reviews, by using a combination of MeSH (medical subject headings) terms. The exact MeSH terms utilized for the database search were penile neoplasms, lymphatic metastasis, lymph node excision, sentinel lymph node biopsy, minimally invasive surgical procedures. We further narrowed down our search results based on predefined terms related to penile cancer, dynamic sentinel lymph node biopsy, and video endoscopic (laparoscopic and robotic) inguinal lymph node dissection. Specifically, we subdivided minimally invasive management into two categories as dynamic sentinel lymph node biopsy and video endoscopic inguinal lymphadenectomy. To include other pertinent studies, we also conducted a manual review of recent review articles, ongoing clinical trials, and recently published scientific abstracts available in the literature. We also excluded studies not published in English or studies published prior to 2001 (Figure 1).
